# Cryo-EM structure of a conjugative type IV secretion system suggests a molecular switch regulating pilus biogenesis

Kévin Macé [1,3 ✉] & Gabriel Waksman [1,2 ✉]

## Abstract

**Conjugative type IV secretion systems (T4SS) mediate bacterial conjugation, a process that enables the unidirectional exchange of genetic materials between a donor and a recipient bacterial cell. Bacterial conjugation is the primary means by which antibiotic resistance genes spread among bacterial populations (Barlow 2009; Virolle et al, 2020). Conjugative T4SSs form pili: long extracellular filaments that connect with recipient cells. Previously, we solved the cryo-electron microscopy (cryo-EM) structure of a conjugative T4SS. In this article, based on additional data, we present a more complete T4SS cryo-EM structure than that published earlier. Novel structural features include details of the mismatch symmetry within the OMCC, the presence of a fourth VirB8 subunit in the asymmetric unit of both the arches and the inner membrane complex (IMC), and a hydrophobic VirB5 tip in the distal end of the stalk. Additionally, we provide previously undescribed structural insights into the protein VirB10 and identify a novel regulation mechanism of T4SS-mediated pilus biogenesis by this protein, that we believe is a key checkpoint for this process.**

**Keywords** Bacterial Conjugation; Type IV Secretion System; T4SS; Antibiotic Resistance; Cryo-EM
**Subject Categories** Microbiology, Virology & Host Pathogen Interaction; Structural Biology

## Introduction

Conjugative type IV secretion systems (T4SSs) minimally contain 12 proteins, named VirB1-VirB11 and VirD4 (Costa et al, 2021; Costa et al, 2023; Waksman, 2019). VirB1 is a lytic transglycosylase, facilitating T4SS assembly (Chandran Darbari and Waksman, 2015). VirB2 is the pilin subunit located in the inner membrane (IM), which is subsequently extracted from the IM and assembled to form a helical extracellular appendage termed 'the conjugative pilus' (Costa et al, 2016). The recent cryo-electron microscopy (cryo-EM) structure of the conjugative Type 4 Secretion System (T4SS) encoded by the plasmid R388 revealed that the remaining VirB proteins form a large assembly organised into four sub-complexes (Fig. 1): the outer membrane core complex (OMCC), the stalk, the arches and the inner membrane complex (IMC) (Mace et al, 2022). The OMCC is made of TrwH/VirB7, TrwF/VirB9 and TrwE/VirB10 (in this study, we use both the R388 Trw and *Agrobacterium* VirB naming nomenclatures). It is itself composed of two parts: the 14-fold symmetrical O-layer embedded within the outer membrane (OM) and the 16-fold symmetrical I-layer beneath it. Previous structural studies of the OMCC have revealed that the O-layer and I-layer often present a mismatch symmetry (Amin et al, 2021; Durie et al, 2020; Sheedlo et al, 2020), and yet they are both composed of the same proteins (for example, TrwF/VirB9 and TrwE/VirB10). The Stalk is 5-fold symmetrical and is made of TrwJ/VirB5 and TrwI/VirB6, with TrwI/VirB6 embedded into the IM and hypothesised to serve as a platform for pilus subunit recruitment and assembly (Mace et al, 2022). Indeed, TrwI/VirB6 features two key sites: a VirB2-binding site at its base in the IM, and a pilus-assembly site at its top. TrwJ/VirB5 is located at the tip of the stalk, and is expected to be recruited at the pilus tip during the early step of pilus biogenesis. The sixfold symmetrical Arches are observed in the previously published structure to be made of six trimers of the periplasmic domain of TrwG/VirB8 (TrwG/VirB8$_{peri}$) forming a large ring around the Stalk. The sixfold symmetrical IMC is made of six copies of a protomer comprising one copy of TrwM/VirB3, two copies of the TrwK/VirB4 ATPase subunits termed TrwK/VirB4$_{central}$ and TrwK/VirB4$_{outside}$, and in the structure published previously three N-terminal tails of TrwG/VirB8 (TrwG/VirB8$_{tail}$). Six TrwK/VirB4$_{central}$ form a central hexamer linked to the IM by VirB3. TrwK/VirB4$_{outside}$ forms a robust dimer with VirB4$_{central}$ and interacts with the TrwG/VirB8$_{tail}$. TrwD/VirB11 was not part of the complex because detergent solubilisation of the complex results in the dissociation of this ATPase. However, its location under the TrwK/VirB4$_{central}$ hexamer was ascertained by co-evolution and site-directed mutagenesis (Mace et al, 2022). Although a third ATPase, TrwC/VirD4, is known to be part of the IMC at some point during conjugation, no density could be found for it, and therefore, its

[1]Institute of Structural and Molecular Biology, Department of Biological Sciences, Birkbeck College, Malet Street, London WC1E 7HX, UK. [2]Institute of Structural and Molecular Biology, Division of Biosciences, Gower Street, University College London, London WC1E 6BT, UK. [3]Present address: Univ. Rennes, CNRS, Institut de Génétique et Développement de Rennes (IGDR) - UMR6290, 35000 Rennes, France. ✉E-mail: kevin.mace@univ-rennes.fr; g.waksman@bbk.ac.uk

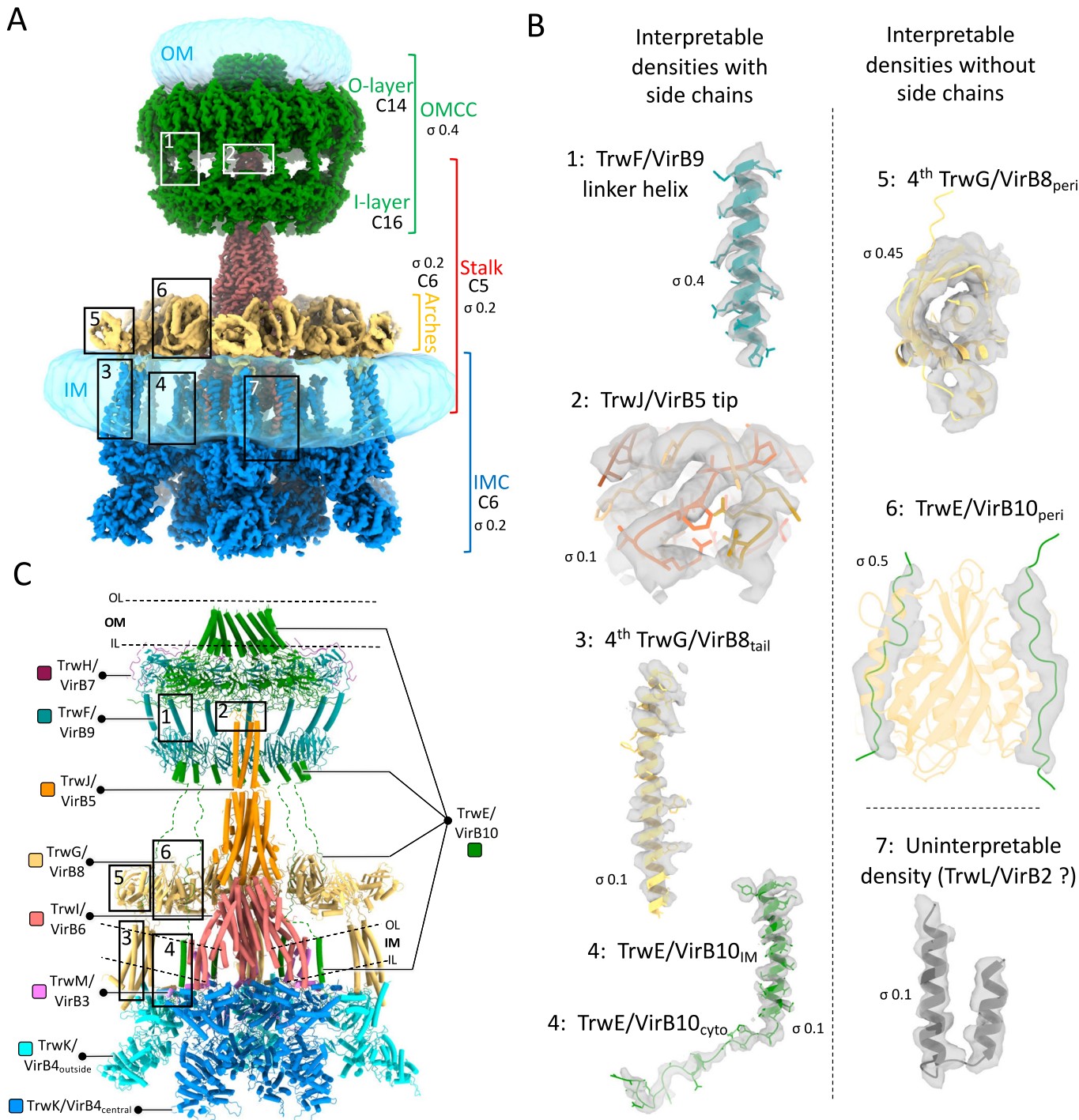

location and interactions with other T4SS components remain elusive.

In the cryo-EM map published previously, certain regions of the complex still presented challenges due to insufficient resolution, preventing a detailed interpretation of some parts of the EM density map. In response to these limitations, we present here an enhanced T4SS structure that builds upon our previous work (Table EV1; Figs. EV1 and EV2). This improved structure was achieved by collecting additional cryo-EM data as well as dedicated image

processing on regions of interest. Specifically, we generated a total of nine cryo-EM maps, with a range of resolutions of 2.5 to 6.2 Å, to gain additional structural insights. These include the elucidation of several conformations of the OMCC, the structure and interactions of the TrwE/VirB10 N-terminus comprising periplasmic, inner membrane, and cytoplasmic regions, the tip region of TrwJ/VirB5, a fourth TrwG/VirB8 subunit in the asymmetric unit of the Arches and IMC, and the connector domain of the Arches formed by the collective contribution of all four TrwG/VirB8 subunits. All

◄ **Figure 1.   Improved structure of the R388 conjugative T4SS.**

(A) Composite EM density map. A composite EM density map of the R388 T4SS is presented and created by assembling maps detailed in Table EV1A. In this map, sub-complexes of the T4SS, including the outer membrane core complex (OMCC; map referred to as "OMCC Conformation A at 3.2 Å"), the Stalk (map referred to as "Stalk C5 at 3.0 Å"), the Arches (map referred to as "Arches at 6.2 Å"), and the inner membrane complex (IMC, map referred to as "extended IMC protomer at 3.8 Å"), are colour-coded in green, red, yellow and blue, respectively. Sigma levels for each map is reported. Symmetry within the sub-complexes is indicated. The detergent and/or lipid densities at the membrane and outer membranes are depicted as semi-transparent light blue density. Additionally, newly identified densities from this study compared to the one published previously are highlighted in rectangular boxes and detailed in panel (B). (B) Newly identified densities. Three categories of newly identified densities are shown as grey, semi-transparent surface contour: interpretable with side chains (regions 1 to 4), interpretable with only secondary structures represented (no side chains; regions 5 and 6) and uninterpretable (region 7). Structures shown in these densities are in cartoon representation, with side chains reported only for regions 1 to 4. The structures are colour-coded according to proteins as in panel (**C**). Sigma levels are reported. For region 7, the density was tentatively ascribed to TrwL/VirB2, but in the absence of corroborating evidence, the density is classed as uninterpretable at this moment in time. (C) Composite model of the T4SS. A composite model of the R388 T4SS is presented in cartoon representation colour-coded per protein as shown in margins. Positions of IM and OM based on cryo-EM densities of detergents and lipids are shown by dashed lines labelled OL and IL for the outer leaflet and inner leaflet of the lipid bilayer. Regions highlighted in (**B**) are shown in correspondingly numbered boxes.

structures are validated by co-evolution and AlphaFold modelling. Altogether, our results greatly enhance our understanding of T4SS molecular mechanisms, notably its regulation of pilus biogenesis by TrwE/VirB10.

## Results and discussion

In the higher resolution cryo-EM maps presented here (Fig. 1A), additional densities were observed, that fall into three categories (Fig. 1B): (i) densities with clear secondary structural features and side chains, where accurate models could be built de novo (regions 1 to 4 in Fig. 1C), (ii) densities with clear secondary structural elements but no side chains, where the main chain could be docked accurately, but no side chains could be built (regions 5 and 6 in Fig. 1C) and finally (iii) insufficiently resolved densities which could not be interpreted (region 7 in Fig. 1C). The latter showed two tubes of densities in the IM that could possibly correspond to TrwL/VirB2 known to contain two TMs, but this interpretation could not be validated and therefore will not be discussed further here.

### The outer membrane core complex (OMCC)

All resolved OMCC structures consistently exhibit a conserved global organisation composed of an O-layer and I-layer sub-complex, often displaying a mismatched symmetry between them (Amin et al, 2021; Durie et al, 2020; Sheedlo et al, 2020). For instance, the OMCC of R388 has a C14/C16 mismatch between the O- and I-layer, respectively (Mace et al, 2022) (Fig. 2). Intriguingly, O- and I-layers are formed of the same proteins: to only mention R388, the O-layer is formed of the CTD of both TrwF/virB9 and TrwE/VirB10, while the I-layer is formed of the NTD of these same proteins. Unsymmetrised C1 maps for the entire OMCCs are obtainable, however, they are often of insufficient resolution to derive important structural features such as the TrwF/VirB9 linker sequence between the two layers or to derive detailed information on the symmetry mismatch. In this study, thanks to a larger dataset, we were able to achieve high resolution without imposing any symmetry constraints (Fig. EV1), enabling a more in-depth investigation of the OMCC.

Initially, we solved the OMCC structure without imposing symmetry at 3.1 Å resolution (Fig. 2A and EV1). In this map, the two additional I-layer binary complexes of TrwF/VirB9$_{NTD}$ and TrwE/VirB10$_{NTD}$ are inserted in diametrically opposite locations (Fig. 2A). Clear density is seen for 14 linker sequences (residues 128 to 150) between the two domains of TrwF/VirB9, which we can now confirm is mostly α-helical. As anticipated, the biggest impact of domain insertion in the I-layer is observed in this linker helix. When the angle it makes relative to a vertical axis is measured, a pattern can be seen, with the largest angles observed on each side of the insertion (see complexes 2 and 10 or 8 and 16 in Fig. 2A, lower panels). The angle tappers off the further the complex is from both insertions (see complexes 4 and 12 in the same Figure panels). Clearly, linker helix flexibility is crucial to accommodate complex insertion in the I-layer.

Further analysis employing 3D classification (Fig. EV1) reveals a degree of heterogeneity in the positioning of the two extra TrwF/VirB9$_{NTD}$-TrwE/VirB10$_{NTD}$ sub-complexes within the I-layer (Fig. 2B). We obtained three distinct OMCC structures, each characterised by a unique arrangement of the two extra sub-complexes in the I-layer. These three arrangements are: (i) "conformation A" which is the one observed previously (see above) with the extra sub-complexes inserted diametrically opposite of each other; (ii) 'conformation B' where insertion of the second extra sub-complex in the I-layer is shifted by 1 compared to conformation A; in this conformation, there are 6 and 8 TrwF/VirB9$_{NTD}$-TrwE/VirB10$_{NTD}$ sub-complexes on each side of the extra sub-complexes; (iii) 'conformation C' where insertion of the second extra sub-complex is shifted by 2 compared to conformation A; in this conformation, 5 and 9 TrwF/VirB9$_{NTD}$-TrwE/VirB10$_{NTD}$ complexes are observed on each side of the extra sub-complexes. Aligning the structures of these three I-layer conformations using the O-layer as the reference underscores how the variability in the positions of the two extra I-layer complexes impacts the overall I-layer structure (Fig. 2C). Although the structure of each complex forming the I-layer do not change significantly (RMSD of 1.06 Å), a shift between corresponding complexes in the superposition of conformations A and B or A and C is observed. In both, this shift changes in magnitude, being maximal in the superposition involving the second extra sub-complex (that in orange in Fig. 2C i.e. complex 10 in the A versus B superposition and complex 11 in the A versus C superposition) and, intriguingly, being minimal in the complexes opposite (complex 2 in the A versus B superposition and complex 3 in the A versus C superposition). Thus, the insertion of extra complexes occurs at various positions in the I-layer,

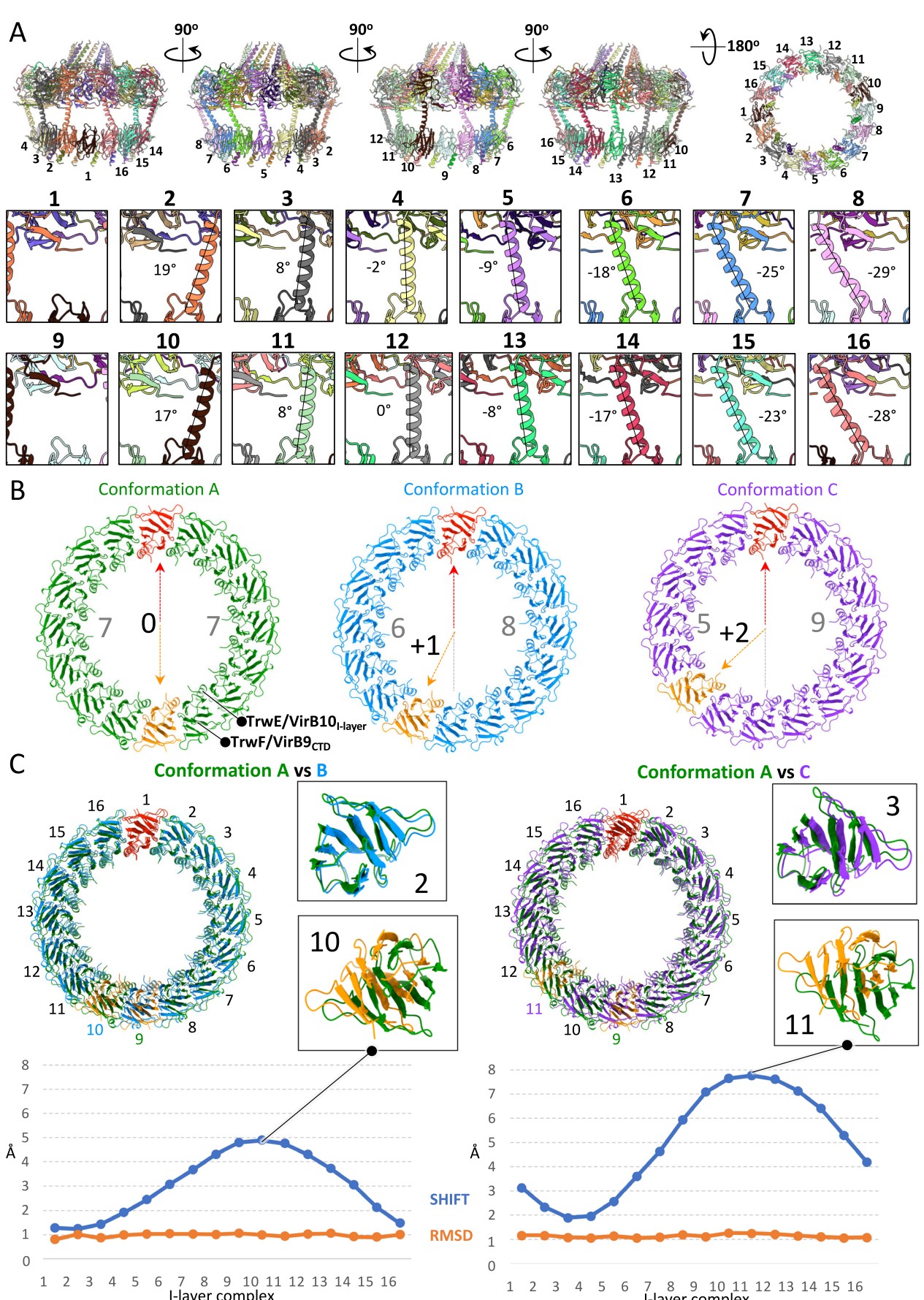

◄ **Figure 2. Analysis of OMCC mismatch symmetry.**

(A) Structure analysis of the OMCC conformation A. In the top panel, five views of the OMCC structure obtained without symmetry applied (from the "Conformation A at 3.2 Å" map) are shown in cartoon representation and coloured by chains. Each I-layer complex is numbered. The bottom panel provides a close-up view of all TrwF/VirB9 linker helices connecting the O-layer to the I-layer, with the angles of the helices relative to the vertical axis indicated. (B) Three distinct conformations of the OMCC. In this panel, three OMCC structures are presented, each exhibiting a distinct I-layer organisation. A top view of the three I-layer structures is shown in cartoon representation, with colours representing different conformations: green for the conformation A, blue for the conformation B and purple for the conformation C. For each conformation, the two extra I-layer complexes are coloured in red and orange. The number of I-layer complexes between the two inserted extra I-layer complexes is indicated in grey. (C) Superposition of OMCC structures. This panel shows pairwise superpositions of the OMCC structures presented in panel (B), with alignment based on the O-layer. Left: superposition of Conformations A and B. Right: superposition of Conformations A and C. The impact of the insertion of two extra complexes in the I-layer is shown at the right of each superposition in zoom-in panels illustrating the least and most affected complexes (2 and 10 for Conformation A versus B and 3 and 11 for Conformation A versus C). Also reported underneath is the shift (as reported using Chimera) separating the two superposed structures for each I-layer complex position. Despite these shifts, each I-layer complex structure remains identical, as evidenced by the RSMD values (orange line). Raw data are included in the source data file. Source data are available online for this figure.

resulting in structural adjustments that propagate over the entire I-layer. The conservation of this mismatch symmetry across various T4SS types emphasises its importance (Amin et al, 2021; Durie et al, 2020; Sheedlo et al, 2020). This distinct pattern observed in the OMCC, characterised by a quasi-symmetrical mismatch, suggests the existence of a specific maturation process during OMCC assembly, although the exact mechanism remains enigmatic. We would like to suggest a plausible role for this asymmetry: to expand the dimensions of the I-layer structure, thereby achieving the ideal diameter to accommodate the growing pilus at a later stage.

## The stalk and arches

In these regions of the T4SS structure, improved resolution resulted in better-defined densities at the tip of TrwJ/VirB5. Also, a fourth subunit of TrwG/VirB8 in the asymmetric unit of both the Arches (TrwG/VirB8$_{peri}$) and the IMC (TrwG/VirB8$_{tail}$) could be placed and built.

In the TrwJ/VirB5 N-terminal tip (Fig. 3A), 11 residues could be added at the N-terminus (residues Gln23-Ala33). Gln23 is the very N-terminal residue in the protein after signal sequence cleavage. As shown in Fig. 3A, middle panel, this newly built region is hydrophobic in nature. VirB5 homologues have been shown to be located at the tip of the pilus, an ideal location to interact with the recipient cell surface (Aly and Baron, 2007). Viewed from the top, the hydrophobic tip of TrwJ/VirB5 resembles a needle (Fig. 3A, insets of each panel). It is not known whether VirB5 proteins can penetrate membranes. However, a body of structural work on this family of proteins ranging from their structural similarities with hemolysin E (Mace et al, 2022) and the hydrophobic nature of their N-terminal tip appear to point to a role in recipient membrane recognition and possibly puncturing.

In our previous report (Mace et al, 2022), the resolution for the Arches was low due to the considerable flexibility of the region. Thus, only three TrwG/VirB8 periplasmic domains (TrwG/VirB8$_{peri}$) in the asymmetric unit (18 total) could be located unambiguously. However, density was observed on the side of the TrwG/VirB8$_{peri}$ ring, a density that could not be interpreted at the time. Additional data collection has, however, resulted here in improved density in this region, allowing us to ascribe it unambiguously to a fourth TrwG/VirB8$_{peri}$ domain. As shown in Fig. 3B, this density is shaped like a hook and protrudes outwards away from the Arches ring. In the hook, one molecule of TrwG/

VirB8$_{peri}$ could be fitted based on secondary structures (see Fig. 1B, region number 5, and Fig. 3B, right-most panel). The resolution was, however, too low to build side chains in this region. Correspondingly, a fourth TrwG/VirB8$_{tail}$ is observed in the asymmetric unit of the IMC (Fig. 3C,D). In this region of the IMC, side chains are clearly visible and a complete model of the 4th TrwG/VirB8$_{tail}$ was built. AlphaFold (Jumper et al, 2021) also predicts a four-helix bundle between the four TrwG/VirB8$_{tail}$ and, reassuringly, this AlphaFold model superimposes well with the one built in the improved density (Fig. 3C,D). Therefore, overall, there are 24 TrwG/VirB8 subunits in the T4SS.

The three TrwG/VirB8$_{peri}$ domains in the asymmetric unit of the Arches described previously were related to each other in a way seen in two different earlier publications: MolA and MolB forms an interface similar to that observed in *Helicobacter pylori* CagV/VirB8$_{peri}$ while the MolB and MolC interface is similar to that observed in *Brucella suis* VirB8$_{peri}$ (Terradot et al, 2005; Wu et al, 2019). The newly observed MolD makes only very few contacts with this trimeric unit and is not related by symmetry with any of the other 3 subunits. However, it appears stabilised (and therefore visible in the density) mostly via interactions of its connector sequence (between TrwG/VirB8$_{peri}$ and TrwG/VirB8$_{tail}$, residues 63 to 94) with the corresponding connector sequences of the other three subunits (Fig. 3C–E). Each connector sequence forms a β-hairpin, the four of them assembling to form an eight-stranded β-sandwich (Fig. 3C–E).

Unexpectedly, two additional densities were observed running along the long axis of TrwG/VirB8$_{peri}$ MolB and MolC (in green in Fig. 4A). These densities showed no side chains and, therefore, could not be unambiguously assigned. Nevertheless, co-evolution analysis and AlphaFold modelling (Figs. 4B and EV3; Dataset EV1) led us to hypothesise that these densities correspond to the sequence of TrwE/VirB10 between residues 83 and 101, a region we label TrwE/VirB10$_{Arches}$. This region interacts with residues in the α-helical side of TrwG/VirB8$_{peri}$ (α4–α6) (Fig. 4C). Previous studies have described unidentified sequences of VirB10 to interact with VirB8$_{peri}$ in the α4-α5 and the β1 regions (Sharifahmadian et al, 2017). In the structure of the fully-assembled T4SS presented here and earlier (Mace et al, 2022), the β1 region is implicated in intra- and inter-asymmetric unit interactions (Fig. 4C). Therefore β1 is unavailable for binding to VirB10. However, as shown in Fig. 4D in red, the α4-α5 region mapped previously (Sharifahmadian et al, 2017) to interact with VirB10 is exactly where we observed the TrwE/VirB10$_{Arches}$ density. Therefore, it is indeed likely that the

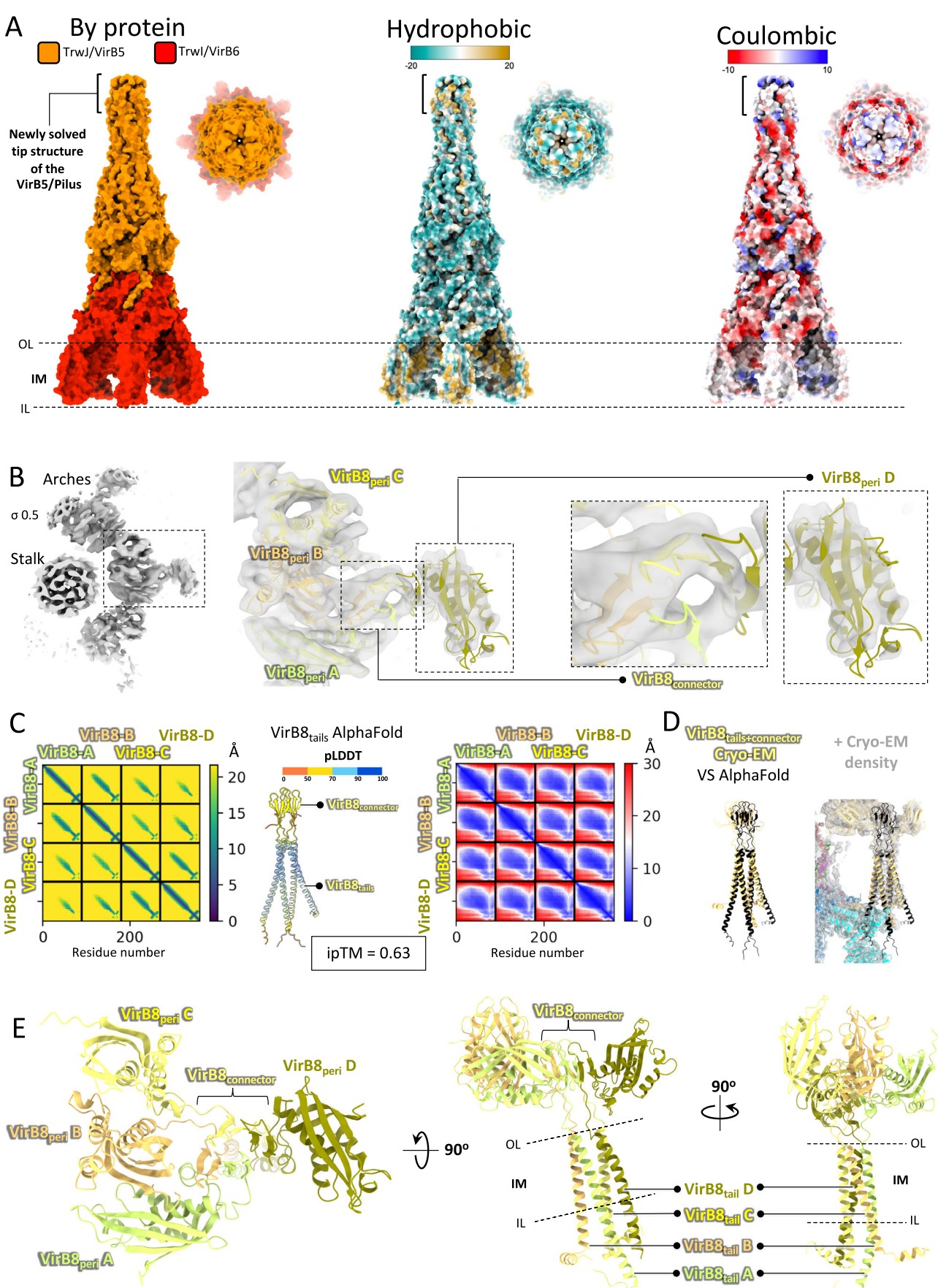

**Figure 3.  Structure of the Stalk and Arches sub-complex.**

(A) The stalk in various representations. In this panel, the Stalk structure is presented in both side and top views, employing three different colour-coding schemes: by proteins (left), hydrophobicity (middle), and coulombic properties (right). The newly resolved tip of the TrwJ/VirB5 structures is indicated in brackets, and the IM is delineated by two dashed lines labelled OL and IL. (B) Details of the cryo-EM map in the Arches region. Left: top view of the "Arches at 6.2 Å" map (Table EV1a) is displayed in grey colour and shows three asymmetric units, one of them (in dashed line box) is better defined. Sigma level is indicated. Middle panel: close-up of the asymmetric unit of the "Arches at 6.2 Å" map corresponding to the region shown in the inset in the left panel. The structure of four TrwG/VirB8$_{peri}$ and four TrwG/VirB8$_{connector}$ regions are shown in cartoon representation (rigid body fitting correlation coefficient of 0.88 and 0.75 for TrwG/VirB8$_{peri}$ and TrwG/VirB8$_{connector}$, respectively). Density is shown in semi-transparent grey. Two dashed rectangles highlight two newly solved portions of the structure compared to our earlier work: the TrwG/VirB8$_{connector}$ and the fourth TrwG/VirB8$_{peri}$, which are further detailed in the zoomed-in views on the right. (C) TrwG/VirB8 Arches structure validation using AlphaFold. Distogram plot (left), predicted model colour-coded by model quality (pLDDT; middle) and PAE (predicted aligned error) plot (right) for the four TrwG/VirB8 tail and connector domains. Interacting residue pairs detected by AlphaFold are marked with green dots on the diagram at left. The ipTM score is reported. A list of co-evolving residue pairs and top-scoring pair mapping onto AlphaFold models are reported in Dataset EV1 and Fig. EV3. (D) Superposition of the AlphaFold (in black ribbon) and cryo-EM (in yellow ribbon) models for the TrwG/VirB8 connector and tail domains (RMSD of 9.398 and 6.928 Å for TrwG/VirB8$_{connector}$ and TrwG/VirB8$_{tails}$, respectively) without (left) or with (right) the cryo-EM EM density map for this region contoured at same sigma level as in panel (B). (E) Structure of the asymmetric unit of the TrwG/VirB8 Arches structure. The four TrwG/VirB8 subunits are colour-coded in four shades of yellow and in cartoon representation. TrwG/VirB8 subunits and domains are indicated, as well as the location of the IM represented by two dashed lines labelled OL and IL.

densities observed belong to TrwE/VirB10 and the assignment based on AlphaFold and co-evolution is correct. No TrwE/VirB10$_{Arches}$ sequences were found bound to TrwG/VirB8$_{peri}$ MolA or MolD. Interestingly, although there are 16 copies of the TrwE/VirB10$_{Arches}$ sequence available in the T4SS structure for binding to VirB8$_{peri}$, only 12 would bind to the VirB8$_{peri}$ Arches region (2 per asymmetric unit in a region that is sixfold symmetrical).

Because of the twofold symmetrical arrangement of MolB and MolC, their TrwE/VirB10$_{Arches}$-binding sites are located in opposite faces of the Arches ring, one facing outwards and the other inwards (Fig. 4C). Since the inner face of the ring surrounds the VirB2 assembly site on VirB6, this may imply that 6 TrwE/VirB10$_{Arches}$ sequences may contact the pilus in the early phase of pilus biogenesis, potentially providing a sensing or checkpoint mechanism for pilus assembly.

## The inner membrane complex (IMC)

From our previous work (Mace et al, 2022), the IMC is a hexamer of an IMC protomer which we defined as including two TrwK/VirB4 subunits (TrwK/VirB4$_{central}$ and TrwK/VirB4$_{outside}$), one TrwM/VirB3 subunit and three TrwG/VirB8$_{tails}$. We now define an "extended IMC protomer" containing not only the same set of proteins but also three additional polypeptides: the fourth TrwG/VirB8$_{tail}$ mentioned above, the TM region (α1 and α2) of TrwI/VirB6 (residues 30–82), and a newly discovered density, corresponding to TrwE/VirB10 (residues 21 to 69) (in green in Fig. 5A). This density runs along TrwI/VirB6 α1 and α2 through the IM (Fig. 5B, inset at right), and then extends in the cytoplasm to make limited contacts with the TrwK/VirB4$_{central}$ subunit of one protomer (TrwK/VirB4$_{central -A}$ in Fig. 5B, inset at right), followed (as we progress towards its N-terminus) by more extensive contacts with the TrwK/VirB4$_{central}$, TrwM/VirB3 and finally the TrwK/VirB4$_{outside}$ of the adjacent protomer (labelled TrwK/VirB4$_{central -B}$, TrwM/VirB3$_{-B}$ and TrwK/VirB4$_{outside-B}$ in Fig. 5C, inset at right). The part of TrwE/VirB10 interacting with the two IM TMs of VirB6 will be referred to as "TrwE/VirB10$_{IM}$" (residues 43 to 69) while the part N-terminal to it interacting with cytoplasm-facing IMC protomer components as 'TrwE/VirB10$_{Cyto}$' (residues 21 to 42).

In these regions of VirB10, the density was of sufficiently high resolution to build side chains. Moreover, AlphaFold predictions of both regions (Fig. 5D–G) yield models that superimpose well with

the model built from the EM density alone (Fig. 5E,G; RMSD of 1.06 and 1.85 Å for TrwE/VirB10$_{IM}$ and TrwE/VirB10$_{cyto}$, respectively). Co-evolution data between VirB10 versus VirB3, VirB4 or VirB6 protein families also confirm the interaction regions (Dataset EV1; Fig. EV3). The first 20 amino acids of TrwE/VirB10 remain invisible, possibly due to disorder or high flexibility.

The interaction between TrwE/VirB10$_{IM}$ with TrwI/VirB6 α1 and α2 are particularly interesting because this binding site for VirB10 on VirB6 entirely overlaps with that of VirB2, the pilus subunit. Indeed, in our earlier work, we identified the site of VirB2-binding on VirB6 using co-evolution analysis and site-directed mutagenesis. We defined this site as "the VirB2 recruitment site" (Mace et al, 2022). The co-evolving residues between VirB2 and VirB6 protein families are shown in Fig. 5H, second panel, while the TrwI/VirB6 residues interacting with TrwE/VirB10$_{IM}$ are shown in Fig. 5H, third panel (right-most). As can be seen, the VirB10$_{IM}$-binding site on VirB6 encompasses entirely the VirB2-binding site, and therefore, VirB10$_{IM}$ in this conformation would prevent VirB2 subunits from being recruited to the VirB6 assembly platform. Therefore, we propose that the regulation of pilus biogenesis is controlled at least in part by the interaction between VirB10$_{IM}$ and the TM region of VirB6. This likely explains our inability to observe a pilus in our biochemical and EM work. The structure of the T4SS we have solved, therefore, represents an early, inhibited, assembly state preceding pilus biogenesis. We have recently shown that the presence of recipient cells considerably increases pilus biogenesis in the R388 system (preprint: Vadakkepat et al, 2024). We, therefore, further hypothesise that contacts between donor and recipient cells may be required to release VirB10$_{IM}$ from VirB6, thereby freeing the VirB2 recruitment site and allowing pilus biogenesis to proceed.

From the work presented here emerges a more detailed description of the VirB10 protein. Two sequences have been described earlier: one termed VirB10$_{O-layer}$ (residues 177–395; also known as VirB10$_{CTD}$ (see below)) which together with VirB9$_{CTD}$ and full-length VirB7 forms the O-layer and also forms the T4SS channel through the OM, and another (residue 135–153) referred to as 'VirB10$_{I-layer}$' which forms an α-helix that interacts with VirB9$_{NTD}$ (Chandran et al, 2009; Mace et al, 2022; Sgro et al, 2018). Here, we identify three additional stretches of sequence, VirB10$_{Arches}$ (residues 83–101) that interact with the VirB8$_{peri}$, VirB10$_{IM}$ (residues 43–69), which makes interactions with

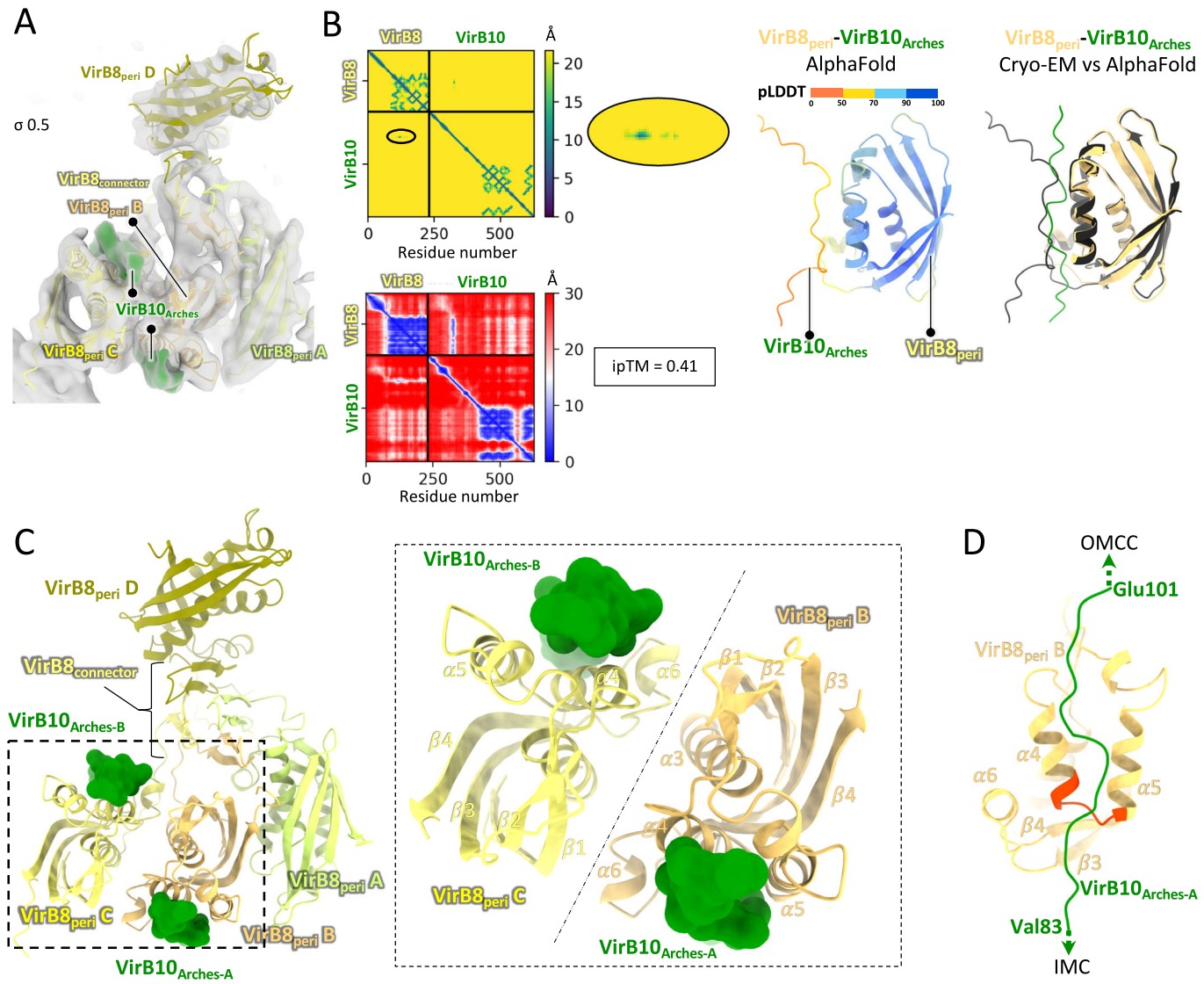

**Figure 4. Structure of the Arches sub-complex and VirB8-VirB10 interaction.**

(A) Top view of the "Arches at 6.2 Å" cryo-EM map showing the two densities corresponding to TrwE/VirB10$_{Arches}$. The map is presented in semi-transparent grey (sigma level indicated). The four TrwG/VirB8 are fitted within this map as represented in Fig. 3, panel E. Two densities corresponding to VirB10$_{Arches}$ are coloured green. (B) Identification TrwE/VirB10$_{Arches}$, the region of TrwE/VirB10 that interacts with TrwG/VirB8$_{peri}$ using AlphaFold. Upper left: distogram plot between VirB10 and VirB8, with interacting residue pairs shown as green dots surrounded by a solid-lined oval. A zoom-up of this region is shown next to it. Lower left: PAE plot with ipTM score. Middle: AlphaFold-derived structural model of TrwG/VirB8$_{peri}$-TrwE/VirB10$_{Arches}$ (residues 88 to 96) shown in cartoon representation, colour-coded by model quality (pLDDT). Right: superposition of the AlphaFold-derived model (in black cartoon) onto the cryo-EM-derived TrwG/VirB8$_{peri}$-TrwE/VirB10$_{Arches}$ structure (in yellow and green cartoon, respectively). RMSD is 0.65 Å. A list of co-evolving residue pairs and top-scoring pair mapping onto AlphaFold models are reported in Dataset EV1 and Fig. EV3. (C) Top view of the structure of the asymmetric unit of the Arches. Left: the four TrwG/VirB8 periplasmic and connector domains are shown in cartoon representation, coloured in four shades of yellow, while the poly-Ala chain of the two TrwE/VirB10$_{Arches}$ are shown in green with a surface representation. A dashed rectangle indicates the zoomed-in area shown on the right. Right: Zoomed-in view of inset shown at left. This view highlights (1) the twofold symmetry between the two TrwG/VirB8$_{peri}$-TrwE/VirB10$_{Arches}$ complexes; (2) the involvement of TrwG/VirB8$_{peri}$ α4 and α5 in binding TrwE/VirB10$_{Arches}$. (D) Details of secondary structures participating in TrwG/VirB8$_{peri}$-TrwE/VirB10$_{Arches}$ interaction. Both proteins are as in panel C, except for the TrwG/VirB8$_{peri}$ region shown in red, which points to a region of VirB8$_{peri}$ shown previously to interact with VirB10 (Sharifahmadian et al, 2017). Secondary structures in TrwG/VirB8$_{peri}$ are labelled, showing interaction in TrwG/VirB8 is principally along the α4 and α5 helices. Residues boundaries for TrwE/VirB10$_{Arches}$ are labelled. Arrows indicate which T4SS sub-complex to which TrwE/VirB10$_{Arches}$ connect, OMCC at the top and IMC at the bottom.

pilus-assembly platform VirB6, and VirB10$_{cyto}$ (residues 21–42) which makes contact with the IMC ATPase complex. Overall, about 72% of the VirB10 sequence is now characterised structurally. A previously noted sequence (residue 101 to 135) includes a proline-rich sequence located between VirB10$_{Arches}$ and VirB10$_{I-layer}$, the

function of which remains unclear (Jakubowski et al, 2009). The positions, boundary residue numbers, and interaction for all regions are summarised in Fig. 6A. One additional fact emerging from this and earlier studies is that VirB10 only contains one folded domain, VirB10$_{CTD}$, functionally overlapping with VirB10$_{O-layer}$

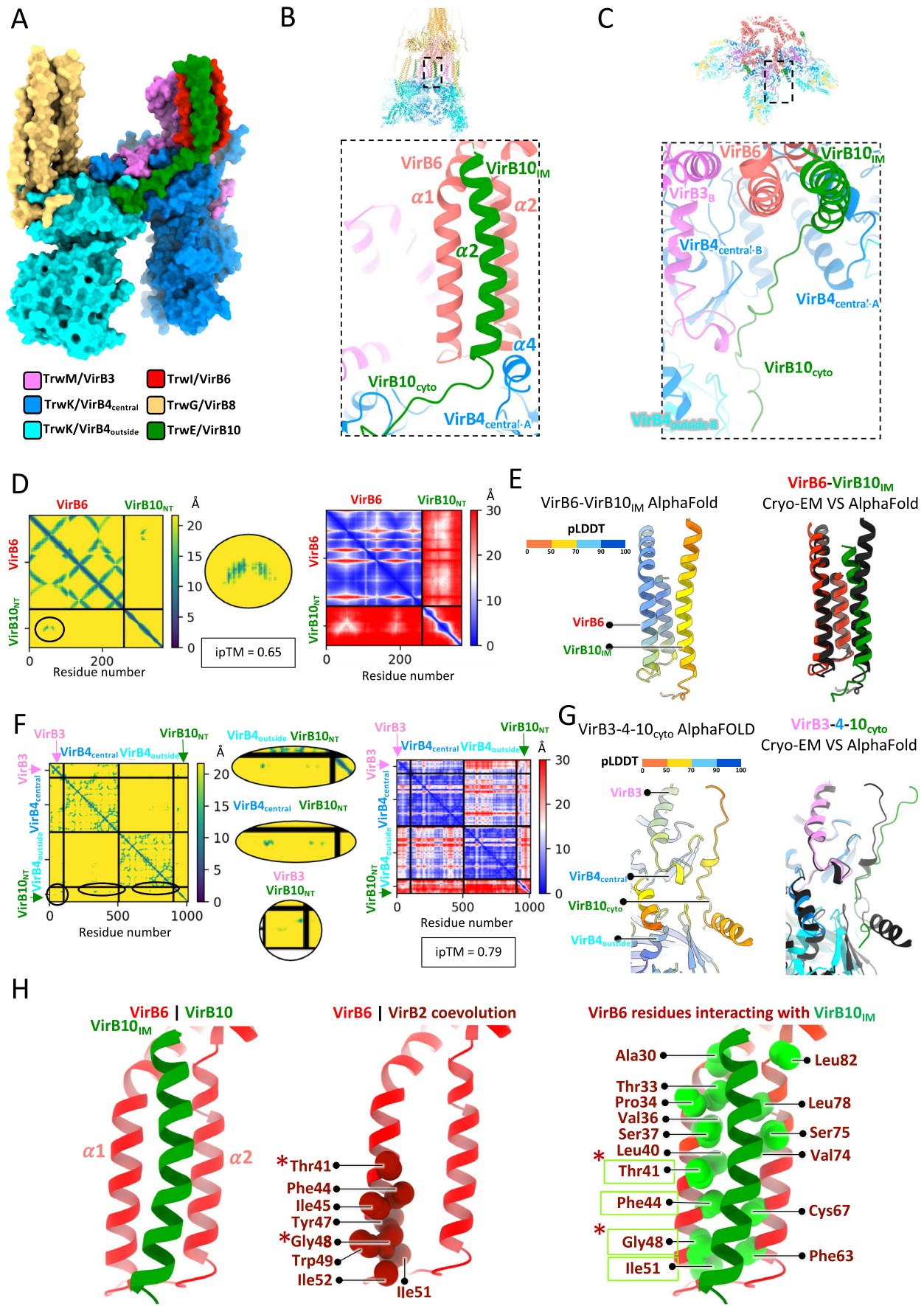

◄  **Figure 5.  TrwE/VirB10 inner membrane and cytoplasmic sub-domains.**

(A) Structure of the extended IMC protomer. This panel presents a side view of the extended IMC protomer structure in surface representation, colour-coded by protein as indicated. The extended IMC protomer includes the formerly-defined IMC protomer (TrwK/VirB4$_{central}$, TrwK/VirB4$_{outside}$, TrwM/VirB3 and four TrwG/VirB8$_{tails}$) to which has been added the TM α1 and α2 of TrwI/VirB6 and TrwE/VirB10$_{IM}$ and TrwE/VirB10$_{Cyto}$. (B) Location and interaction details of TrwE/VirB10$_{IM}$ with the TM helices (α1 and α2) of TrwI/VirB6. Representation and colour coding of proteins are as in Fig. 1C. Inset locates the region of the structure zoomed-in at right. (C) Location and interaction details of TrwE/VirB10$_{Cyto}$ with the TrwK/VirB4$_{central}$-TrwK/VIrB4$_{outside}$-TrwM/VirB3 part of the T4SS complex. Representation and colour coding of proteins are as in Fig. 1C. Inset locates the region of the structure zoomed-in at right. (D) Validation of VirB10-VirB6 interaction using AlphaFold. Left: distogram plot between VirB10$_{NT}$ (see definition of VirB10$_{NT}$ in the main text) and VirB6 with interacting residue pairs shown as green dots surrounded by a solid-lined oval. Middle: a zoom-up of this region is shown. Right: PAE plot. The ipTM score is reported. A list of co-evolving residue pairs and top-scoring pair mapping onto AlphaFold models are reported in Dataset EV1 and Fig. EV3. E Comparison between the cryo-EM and AlphaFold structures of the TrwE/VirB10$_{IM}$-TrwI/VirB6 complex. Left: AlphaFold-derived structural model of the TrwE/VirB10$_{IM}$-TrwI/VirB6 complex shown in cartoon representation, colour-coded by model quality (pLDDT). Right: superposition of the AlphaFold-derived (in black cartoon) and cryo-EM-derived structure of the TrwE/VirB10$_{IM}$-VirB6$_{α1α2TM}$ complex (in green and red cartoon, respectively). RMSD is 1.067 Å. (F) Validation of VirB10 interaction with VirB3 and VirB4 using AlphaFold. Left: distogram plot between VirB10 N-terminus (VirB10$_{NT}$; see definition of VirB10$_{NT}$ in main text), VirB3 and VirB4, with interacting residue pairs involving TrwE/VirB10$_{NT}$ shown here as green dots surrounded by a solid-lined oval. A zoom-up of the various regions of interest is shown in the middle. Right: PAE plot. ipTM score is reported. A list of co-evolving residue pairs and top-scoring pair mapping onto AlphaFold models are reported in Dataset EV1 and Fig. EV3. (G) Comparison between the cryo-EM and AlphaFold structure of TrwE/VirB10$_{Cyto}$-TrwK/VirB4-TrwM/VirB3 complex. Left: Alphafold-derived structural model of the TrwE/VirB10$_{Cyto}$-TrwK/VirB4-TrwM/VirB3 complex, shown in cartoon representation, colour-coded by model quality (pLDDT). Right: superposition of the AlphaFold-derived (in black cartoon) and cryo-EM-derived structures of the TrwE/VirB10$_{Cyto}$-TrwK/VirB4-TrwM/VirB3 complex (in dark and cyan blue cartoon for TrwK/VirB4$_{central}$ and TrwK/VirB4$_{outside}$, respectively, pink cartoon for TrwM/VirB3 and dark green cartoon for TrwE/VirB10$_{Cyto}$). RMSD is 1.85 Å. (H) TrwI/VirB6 α1 and α2 TM interactions with TrwE/VirB10$_{IM}$. Left: structure of the VirB6 α1 and α2 TM interaction with VirB10$_{IM}$ in cartoon representation coloured in red and green, respectively. Middle: TrwI/VirB6 α1 and α2 in cartoon representation except for their residues known to co-evolve with the VirB2 pilus subunit shown as red spheres. Co-evolved residues are labelled, with a star, indicating that the residue was mutated in our previous study and the mutation was shown to result in a significant reduction of T4SS conjugation activities. Right: the same view as in the left panel, with the cα of TrwI/VirB6 residues interacting with TrwE/VirB10$_{IM}$ shown as green balls and labelled. A green box surrounding a residue name indicates that the VirB6 residue co-evolved in both interactions, with VirB2 and VirB10. The star is in the middle panel.

(Fig. 6A). The sequence N-terminal to this domain (residue 1–177) has been often referred to as VirB10$_{NTD}$. However, it does not contain a folded domain, and therefore, it is more accurately named as VirB10 N-terminal sequence or VirB10$_{NT}$ (Fig. 6A). These new insights into VirB10 structure and function provide the means to draw a more complete topology diagram for this protein. It is shown in Fig. 6B, left panel. It provides an update on the naming of the various secondary structures that VirB10 sequences adopt from the N-terminus in the cytoplasm to the C-terminus near the OM.

Zooming out to obtain an overall view of the VirB10 protein within the T4SS structure as shown in Fig. 6B, right panels, it becomes clear that VirB10 can potentially make contact with the pilus all along the pilus length and with its assembly site, suggesting it may play a major role in regulating and/or facilitating pilus biogenesis at many different stages. Not only the two channel-forming α-helices (termed VirB10$_{OM}$ in Fig. 6) would make contact with the pilus as it is threaded through the OM channel, but also a residue of the periplasmic part of the VirB10$_{O-layer}$ has been shown to be potentially involved in the gating mechanism of that channel (Banta et al, 2011; Chandran et al, 2009). Further downstream, the VirB10$_{I-layer}$ could potentially contact the pilus. One of the two VirB10$_{Arches}$ that we have observed runs very close to the VirB2 assembly site on VirB6 and we now know that VirB10$_{IM}$ in the IM obstructs the VirB2 recruitment site on VirB6, providing two other means by which VirB10 may be regulating pilus subunit recruitment and assembly. Finally, we observe a sequence of VirB10 (VirB10$_{Cyto}$) that makes multiple contacts with the TrwK/VirB4 ATPase, thereby potentially providing another point of pilus biogenesis regulation.

An intriguing feature of the T4SS structure solved previously was the paucity of interactions between the various sub-complexes (OMCC, Arches, Stalk and IMC). However, from the work presented here (Fig. 6B), VirB10 emerges as a crucial and unique element that 'glues' these distinct sub-complexes together.

Another remarkable feature of VirB10 is that the T4SS include 16 copies of them, only 14 are used in the O-layer, potentially only 12 of them in the Arches, with only five of those involved in VirB6 binding and six involved in VirB4 ATPase contacts (Fig. 6, middle panel). Understanding such a striking mismatch symmetry and why so many VirB10 are needed when so few may be used is one of the most puzzling features of the T4SS, a puzzle that further structural work will undoubtedly solve.

## Conclusion

The study presented here provides additional insights that shed new lights onto several aspects of T4SS structure and function. We were able to provide further details on the structural dynamics of the OMCC and the various conformations it can adopt. We characterised a hydrophobic tip at the N-terminus of VirB5, and a fourth VirB8 subunits adding to the Arches and the IMC. However, the functional significance of these new structural features remains unclear. More functionally significant, perhaps, has been the characterisation of a VirB10 region that has the potential to interfere with an essential T4SS function which is involved in pilus biogenesis. Indeed, VirB10 proteins define a rather rare class of membrane proteins that spans both the IM and OM of Gram-negative bacteria. Using two-hybrid or phage-display methods as well as biochemical screening of peptide libraries, VirB10 has been shown to interact with most T4SS components (Mary et al, 2018; Sharifahmadian et al, 2017; Terradot et al, 2004). One well-documented interaction of VirB10 is with the VirD4 ATPase, the so-called coupling protein, because it couples substrate recruitment to substrate transfer (Llosa et al, 2003). However, the molecular basis for any of these roles has been unclear. Here, in addition to further details concerning the structure of the entire T4SS, we reveal a new role for VirB10 in pilus biogenesis and provide the molecular basis for this role. Given the considerable importance

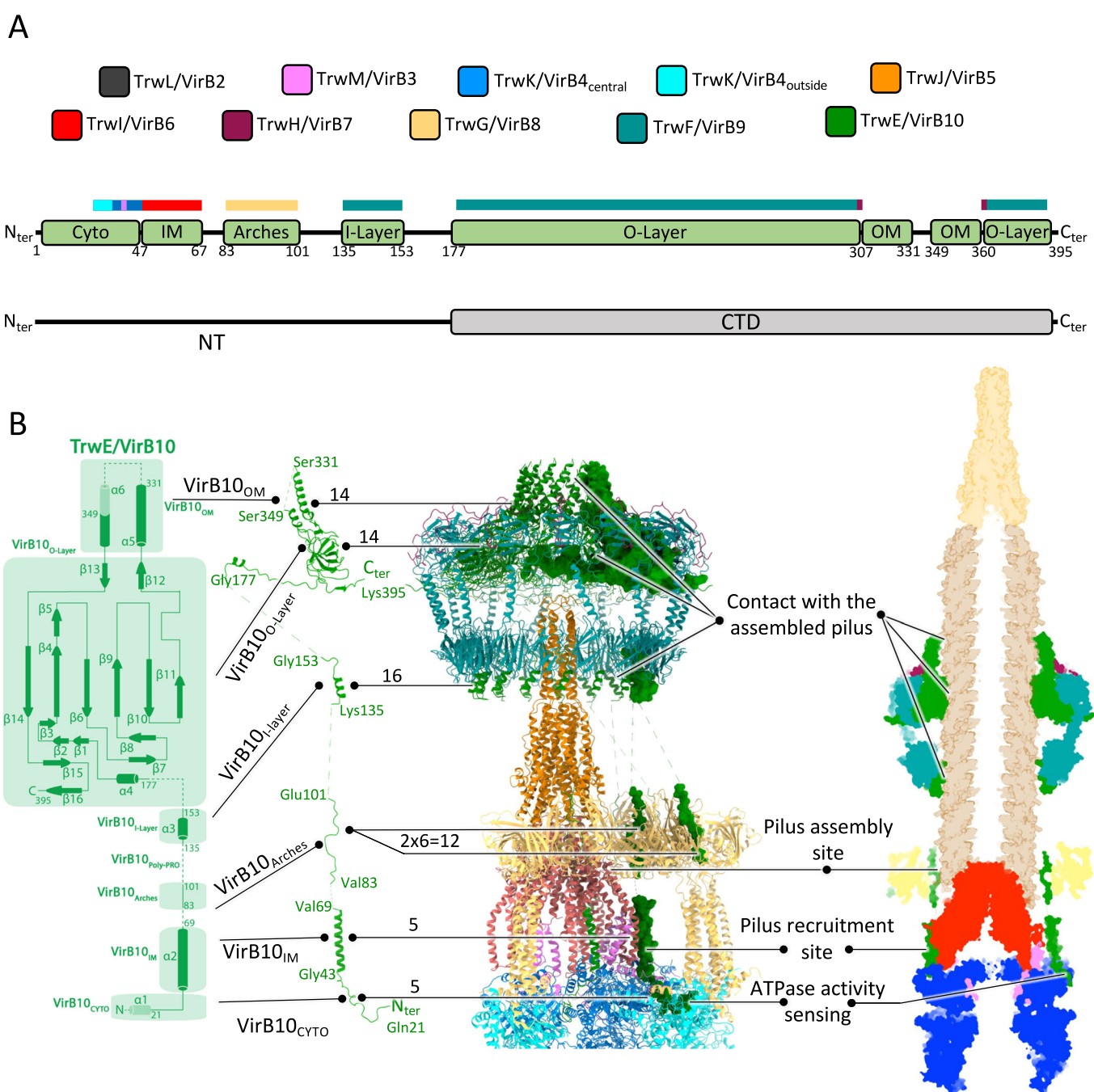

**Figure 6. VirB10: a central protein of the T4SS.**

(A) Structural organisation and interactions of TrwE/VirB10. Top panel, colour-code for Trw/VirB proteins used in this figure. Middle panel: functional organisation of TrwE/VirB10. Green boxes indicate the location and boundary residues of the various functional regions of TrwE/VirB10. In the main text, these functional regions are labelled TrwE/VirB10$_{XXX}$ (for example, TrwE/VirB10$_{Arches}$ or TrwE/VirB10$_{IM}$) where the XXX subscript reflects the function and location of that region. The Trw/VirB proteins with which region TrwE/VirB10XXX interact are indicated on the coloured boxes just above (the colour is by proteins as in the top panel). Bottom panel: folded domain structure of TrwE/VirB10. While two domains were previously described, only one is now known to adopt a defined fold, TrwE/VirB10$_{CTD}$, which is part of the O-layer et forms the OM channel. The previously named NTD is a linear peptide best described as an N-terminal region or TrwE/VirB10$_{NT}$. (B) TrwE/VirB10 full-length structure details. Left: a topology diagram of TrwE/VirB10, with annotated domains and secondary structures. Middle left: The full-length VirB10 structure is presented in cartoon representation. Middle right: The T4SS structure is shown in cartoon representation, except for TrwE/VirB10, which is displayed in surface representation. The number of VirB10 copies known to make interactions with other VirB proteins or itself in the various T4SS regions and sub-complexes is indicated. Right: A cut-out side view in the surface representation of the T4SS-pilus model, with the pilus and its TrwJ/VirB5 tips. The VirB10's potential functions by region/domain during pilus biogenesis are indicated. See main text for details.

that the pilus plays in conjugation, unravelling novel regulatory mechanisms for its biogenesis should provide new approaches targeting their inhibition, potentially leading to the design of new tools to stop the spread of antibiotic-resistance genes.

# Methods

## Bacterial strains, constructs, expression and purification of T4SS

R388 T4SS complexes were expressed and purified from membranes as described previously (Mace et al, 2022).

## Cryo-EM grid preparation and data acquisition

Grids preparation and data acquisition were as described previously (Mace et al, 2022).

## EM processing

MOTIONCOR2 (Zheng et al, 2017) was employed for motion correction and dose weighting, followed by CTF estimation using CTFFIND v4.1 (Rohou and Grigorieff, 2015). EM processing workflows for the various parts of the T4SS are shown in Figs. EV1, EV2.

## Image processing of the T4SS OMCC

### Pre-processing

Reprojections of a low pass filtered (20 Å) map generated using PDB 3JQO (Chandran et al, 2009) were used to pick particles centred on the OMCC with GAUTOMATCH v0.56 (Zhang, 2017). Following multiple rounds of 2D classification, 1,742,107 particles were selected. After 3D heterogeneous classification without symmetry applied, the two best classes were selected, resulting in a selection of 1,360,271 particles that will be used in all subsequent processing. A refined map at 3.12 Å resolution without symmetry applied was then obtained, that served as a reference map in all subsequent processing.

### O-layer and I-layer high resolution

Reference map and particles were imported into RELION 3.1 (Scheres, 2012) for 3D classification without alignment. The best class was used in two rounds of local refinements using CRYOSPARC (Punjani et al, 2017), one focused on the O-layer with C14 symmetry applied, and the other comprising the I-Layer with C16 symmetry applied. The resulting EM density maps had an average resolution of 2.46 Å for the O- Layer (map termed "O-layer C14 at 2.5 Å" in Table EV1) and 2.69 Å for the I-Layer (map termed 'I-layer C16 at 2.7 Å' in Table EV1) as estimated using the gold standard Fourier Shell Correlation (FSC) with a 0.143 threshold.

### OMCC heterogenicity analysis

Reference map and particles were used for heterogeneous refinement without symmetry applied into five classes. After visual inspection in CHIMERA v1.4 (Pettersen et al, 2004), particles from classes showing the same I-layer conformation were combined for a final homogeneous refinement using CRYOSPARC, resulting in three maps: Conformation-A, -B and -C at resolutions of 3,18, 2.93 and 3.05 Å, respectively (Table EV1).

## Image processing of the T4SS Stalk-Arches-IMC

### Pre-processing

Reprojections of the negative-strain EM map of the IMC, Stalk and Arches (EMDB 3585 (Redzej et al, 2017)) were used to pick particles centred on the IMC, Stalk and Arches using GAUTOMATCH. Following multiple rounds of 2D classification using CRYOSPARC, 1,0482,424 particles were selected. After 3D heterogeneous classification without symmetry was applied using CRYOSPARC, the two best classes were selected, resulting in a selection of 600,997 particles and a refinement map at 7.35 Å resolution without symmetry being applied. Both particles and map were used as reference for subsequent analysis.

### Stalk analysis

From the reference map and particles, two rounds of local 3D classification (using the Stalk mask described previously (Mace et al, 2022)) were performed using RELION. The final best class, composed of 104,720 particles was selected for a round of non-uniform refinement (NU-refinement) with C5 symmetry applied using CRYOSPARC, yielding to a 2.97 Å resolution map (termed 'Stalk C5 at 3.0 Å' in Table EV1).

### Arches analysis

From the reference map and particles, a round of local 3D classification was performed using RELION and a previously described mask (Mace et al, 2022). The best class, composed of 115,034 particles, was selected to perform a round of NU-refinement without symmetry applied using CRYOSPARC, yielding a 6.22 Å resolution map (termed "Arches at 6.2 Å" in Table EV1).

### Extended IMC protomer analysis

From the reference map and particles, a round of local 3D classification was performed using RELION and a mask similar to that described in our previous study for the IMC protomer (defined at the time to only contains two TrwK/VirB4, one TrwM/VirB3, and 4 TrwG/VirB8$_{tails}$) but, this time, extended to encompass the TrwE/VirB10 IM and Cyto regions, the TrwI/VirB6 TM region (α1 and α2) and the non-interpretable density shown in Fig. 1, box 7 (Fig. EV2). The best class, composed of 234,578 particles was selected to perform NU-refinement without symmetry, applied using CRYOSPARC, yielding a 3.83 Å resolution map (termed "extended IMC protomer at 3.8 Å" in Tables 1 and EV1).

### Stalk-Arches-IMC analysis

From the reference map and particles, two local 3D classifications were performed using RELION and a mask surrounding the area of interest. The final best class, composed of 65,178 particles, was selected to perform NU-refinement without symmetry applied using RELION, yielding a 4.33 Å resolution map (map termed 'Stalk-Arches-IMC at 4.3 Å').

All maps were subjected to sharpening using DEEPEMHANCER (Sanchez-Garcia et al, 2021) in 'tightTarget' mode, and local resolution estimated using CRYOSPARC. Detailed map statistics are provided in Table EV1.

## Structures model building

### For the OMCC

We employed the previously determined O-layer (PDB-7O3J (Mace et al, 2022)) and I-layer (PDB-7O3T (Mace et al, 2022)) R388 structures as starting models. These models were manually fitted into the new O-layer and I-layer highest resolution maps (those where either C14 or C18 symmetry was applied, respectively (see above)) using COOT v0.9.3 (Emsley and Cowtan, 2004) and refined using PHENIX v1.18.2 (Adams et al, 2010) with secondary structural elements and Ramachandran restraints applied. To generate the OMCC conformations A, B and C models, the O- and I-layer structures were docked into the O- and I-layer densities of the corresponding maps and the linker helix between TrwF/VirB9 CTD and NTD was built and fitted manually (using COOT) into density for all TrwF/VirB9 subunits where the density if visible (14 subunits out of 16). The resulting models were refined using PHENIX, ultimately yielding a final model for Conformation-A, -B and -C OMCC structures (Table EV1).

### For the stalk

We used the previously solved stalk structure (PDB-7O3V (Mace et al, 2022) as a starting model to build a more complete model into the higher resolution map that we describe here ('Stalk C5 at 3.0 Å' map in Table EV1) using COOT. This structure incorporates new elements, such as the TrwJ/VirB5 tip and TrwI/VirB6 transmembrane domains. This model was then refined using PHENIX to generate the final Stalk structure.

### For the arches

We started with the previously solved arches structure (PDB-7OIU (Mace et al, 2022)). To this structure, using the "Arches at 6.2 Å" map, we added in the asymmetric unit of the arches a fourth TrwG/VirB8$_{peri}$ domain and the TrwG/VirB8 connector domain derived from AlphaFold modelling. All the side chains were removed from this model, and further improvements were manually achieved using COOT. PHENIX was used for final refinement (Table EV1).

### For the extended IMC protomer

We began with the previously solved IMC protomer structure (PDB-7Q1V (Mace et al, 2022)). To this structure and using the "extended IMC protomer at 3.8 Å" map, we built a fourth TrwG/VirB8$_{tail}$, the TrwI/VirB6$_{TM}$ helices (α1-α2), and the VirB10$_{IM}$ and VirB10$_{cyto}$ regions. This initial model underwent manual improvements using COOT and final refinement using PHENIX, leading to the extended IMC protomer structure containing not only the two TrwK/VirB4 subunits, the TrwM/VirB3 subunit and the four TrwG/VirB8$_{tails}$, but also the TrwI/VirB6 TM region as well as the TrwE/VirB10 IM and cyto regions (Table EV1).

### For the stalk-arches-IMC

The structural model was constructed using the stalk-arches-IMC at 4.3 Å map and the high-resolution solved structures of the stalk, arches asymmetric unit and extended IMC protomer. This map was important to characterise contacts of TrwE/VirB10$_{CYTO}$ that are shared with two adjacent TrwK/VirB4 subunits as shown in Fig. 5C, right panel. In this composite model, regions with poor Cα backbone density were removed as well as all side chains. COOT was employed for manual improvement, and PHENIX was used for final refinement to generate the Stalk-Arches-IMC structure (Table EV1).

Quality assessments of all structures were performed using MolProbity v4.5.1 (Davis et al, 2007). Data and model statistics are provided in Table EV1B.

EM maps and atomic models were deposited to the EMDB and PDB data bases. Accession codes can be found in data availability section and in Table EV1.

### Structure validation using AlphaFold

To validate the new structural features and interactions presented in this study, we utilised AlphaFold (Jumper et al, 2021) through the ColabFold advanced notebook (Mirdita et al, 2022). This version of ColabFold generates various outputs, including distograms, PAE plots and ipTM scores (presented in Figs. 3–5) and lists of co-evolution pairs with their scores (presented in Dataset EV1; Fig. EV3).

### Interaction analysis and representation of the T4SS structure

Interaction analysis was carried out using the PISA server (Krissinel and Henrick, 2007), and structure figures were generated using ChimeraX v1.6 (Pettersen et al, 2021).

## Data availability

All EM maps and associated atomic models have been deposited in the EMDB and the PDB, respectively. The accession codes are PDB 8RT4 (EMD-19478) and PDB 8RT5 (EMD-19479) for the locally and symmetrised refined structures of O-layer and I-layer respectively, PDB 8RT6 (EMD-19480), PDB 8RT7 (EMD-19481) and PDB 8RT8 (EMD-19482) for the locally and non-symmetrised structures of the OMCC conformation A, conformation B and conformation C respectively, PDB 8RT9 (EMD-19483) for the locally and symmetrised refined structure of the Stalk, PDB 8RTA (EMD-19484) for the locally refined structure of the Arches asymmetric unit, PDB 8RTB (EMD-19485) for the locally refined structure of the extended IMC protomer, and PDB 8RTD (EMD-19488) for the locally and non-symmetrised structure of the overall Stalk-Arches-IMC complex.

The source data of this paper are collected in the following database record: biostudies:S-SCDT-10_1038-S44318-024-00135-z.

## Peer review information

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

## Acknowledgements

We would like to thank Dr. David Houldershaw for IT support and Natasha Lukoyanova for the preparation of EM grids and collection of data. This work was supported by Wellcome grants 098302 and 217089 to GW. Cryo-EM data for this investigation were collected at the ISMB EM facility at Birkbeck

College, University of London with financial support from Wellcome Trust (202679/Z/16/Z and 206166/Z/17/Z). KM purified the T4SS complex. EM processing and AlphaFOLD analysis work was carried out by KM and KM built and refined the structure. KM and GW wrote the manuscript. GW supervised the work.

## Author contributions

**Kévin Macé**: Formal analysis; Investigation; Methodology; Writing—original draft; Writing—review and editing. **Gabriel Waksman**: Supervision; Funding acquisition; Writing—original draft; Project administration; Writing—review and editing.

Source data underlying figure panels in this paper may have individual authorship assigned. Where available, figure panel/source data authorship is listed in the following database record: biostudies:S-SCDT-10_1038-S44318-024-00135-z.

## Disclosure and competing interests statement

Gabriel Waksman is an editorial advisory board member at the EMBO Journal. This has no bearing on the editorial consideration of this article for publication.

# Expanded View Figures

**Figure EV1. Summary of maps and structure and workflow for OMCC maps.**

(A) Summary of maps and structures. Top: unsharpened cryo-EM maps displayed. Sigma level at which the maps have been contoured are reported. Bottom: structures derived from maps and colour-coded as in Fig. 1. "Side chains" and "secondary structures" labels indicate which structures are reported with side and main chains, respectively. (B) Image processing workflow for the obtained OMCC maps. This panel provides details about the image processing workflow for the OMCC maps. It reports on all steps during processing Final sharpened maps coloured by local resolution are shown. FSC curve and angle distribution are reported.

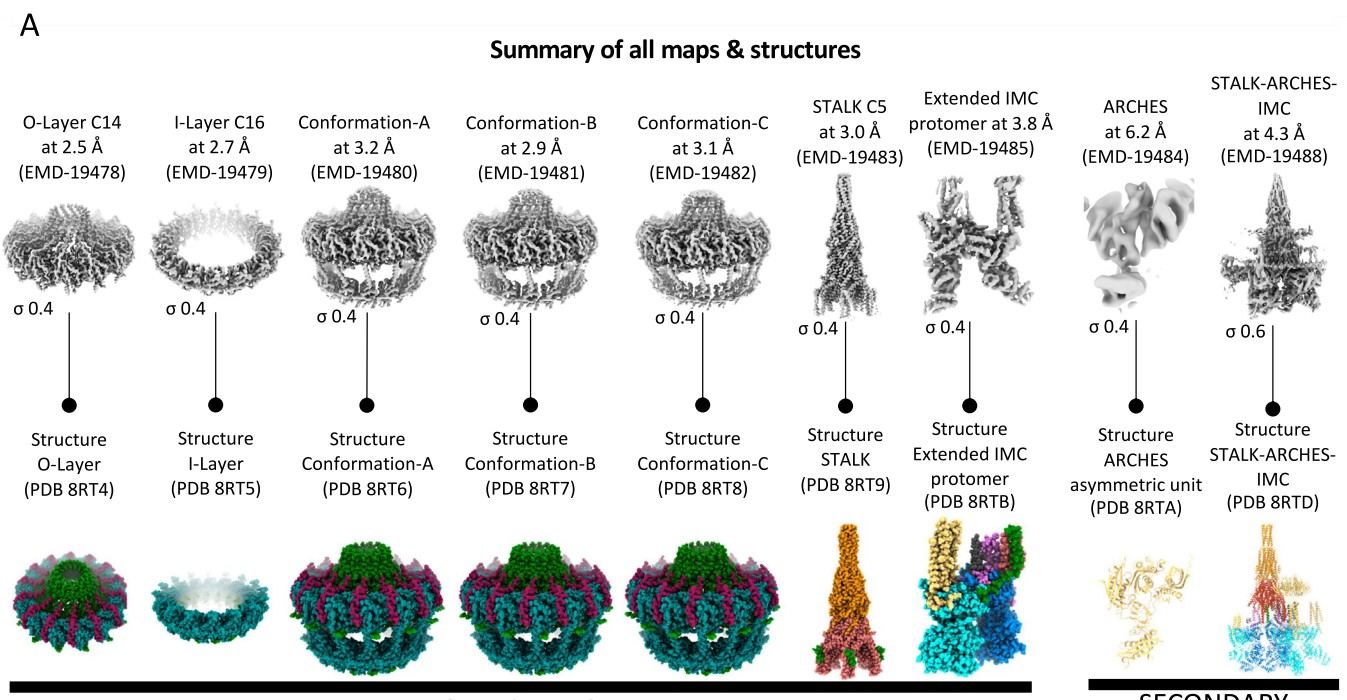

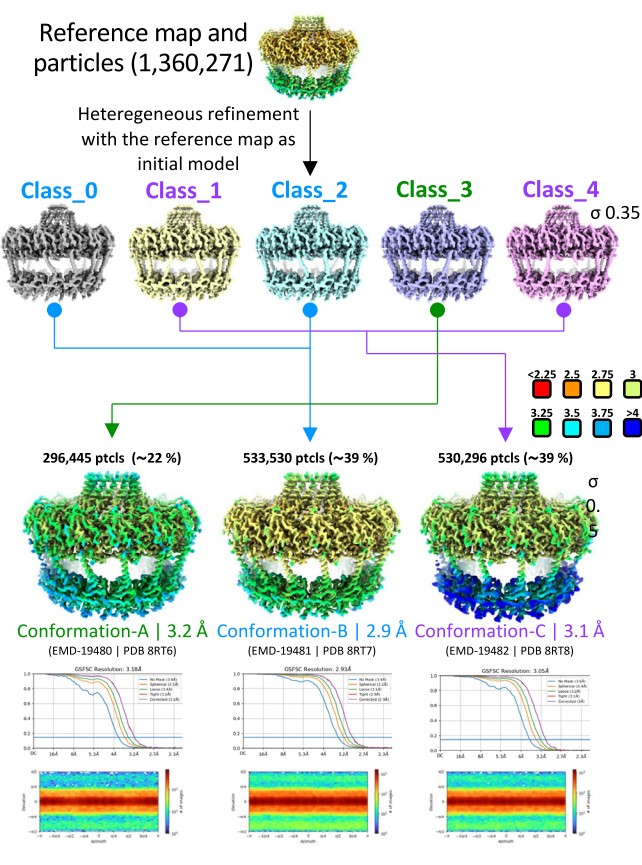

**Stalk image processing**

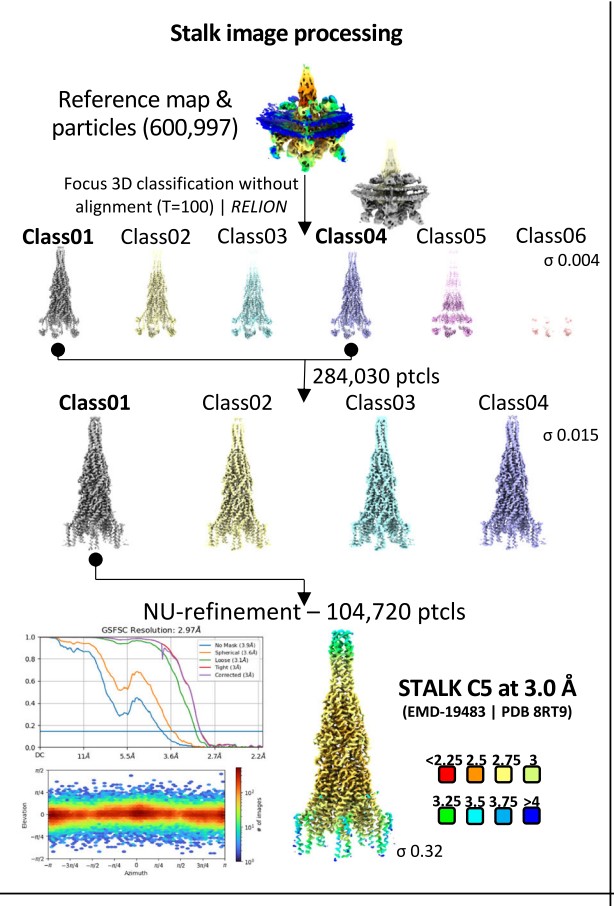

**Arches image processing**

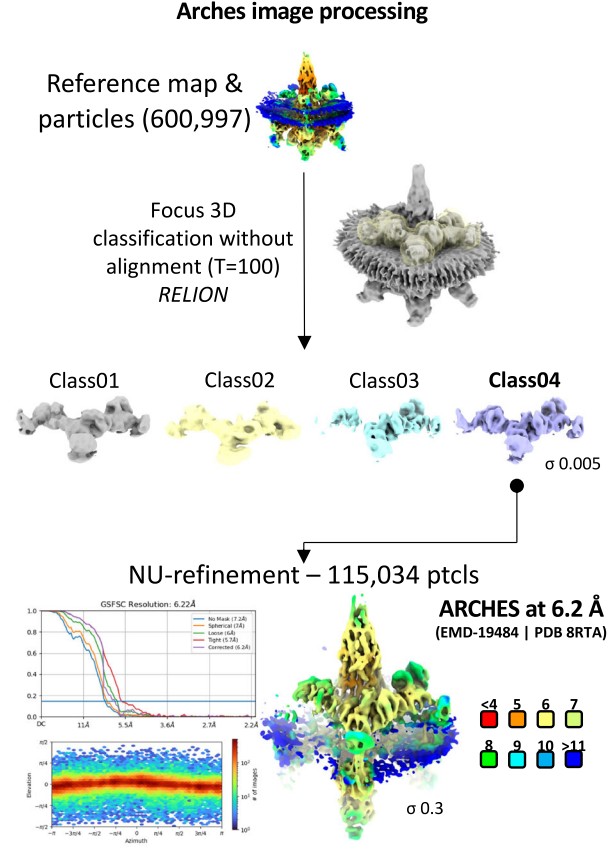

**Extended IMC protomer image processing**

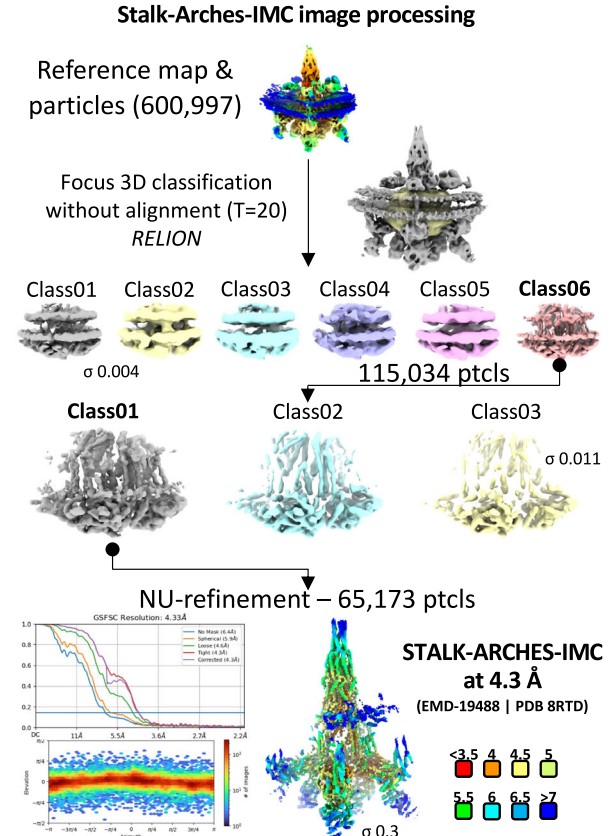

**Stalk-Arches-IMC image processing**

**Figure EV2.   Image processing workflow for stalk, arches asymmetric unit, extended IMC protomer and stalk-arches-IMC.**

This figure provides a detailed overview of the image processing workflow for the stalk, arches asymmetric unit, extended IMC protomer and stalk-arches-IMC cryo-EM maps. It reports on all steps during processing. Final sharpened maps coloured by local resolution are shown. FSC curve and angle distribution are reported.

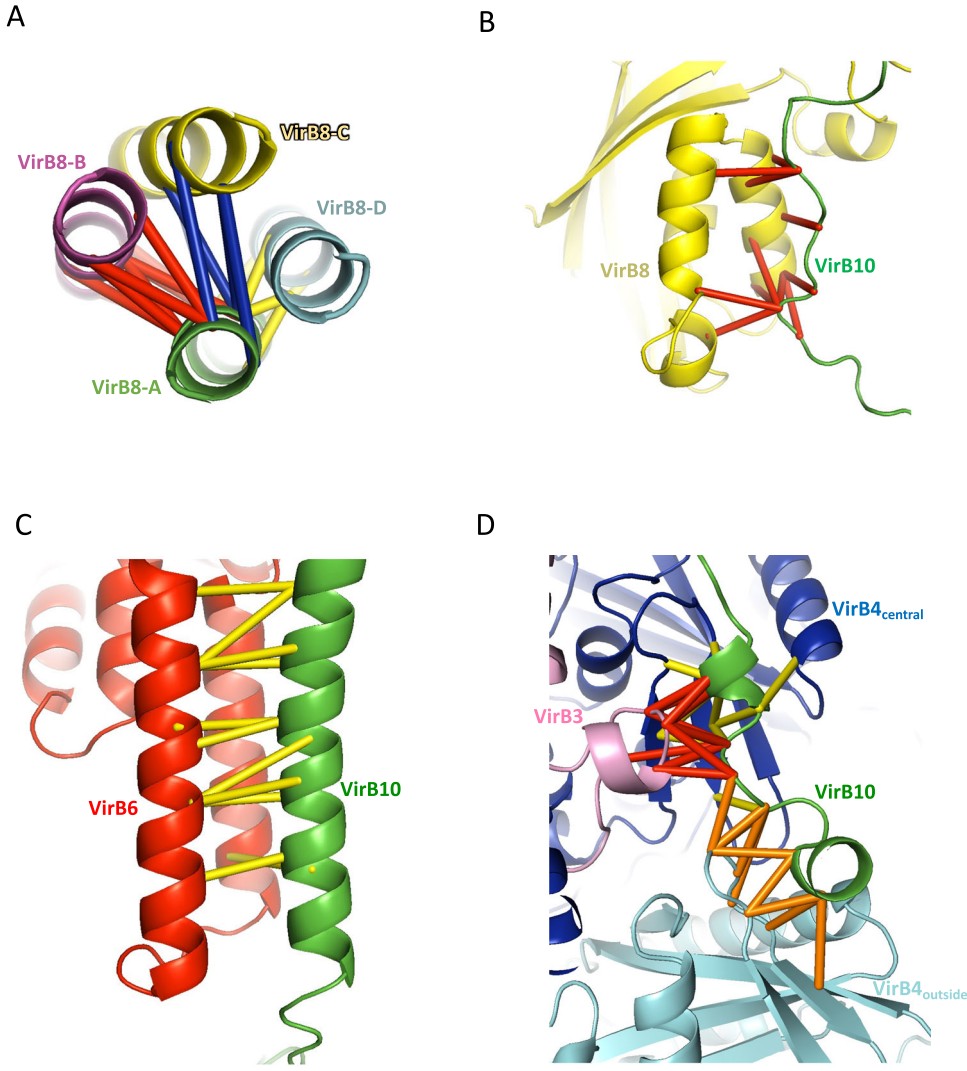

**Figure EV3.  AlphaFold models with top-scoring co-evolving residue pairs mapped.**

(A) Co-evolving residues at the interface of VirB8-A with VirB8-B, VirB8-C and VirB8-D. The ten top-scoring residue pairs listed in Dataset EV1 are mapped for each interaction onto the corresponding Alphafold model. Colour coding for proteins is green, purple, yellow, and cyan blue for VirB8-A, VirB8-B, VirB8-C and VirB8-D, respectively. Residue pairs are shown by bars between their Calpha atoms, colour-coded red, blue, and yellow for VirB8-A and B, VirB8-A and C and VirB8-A and D interactions, respectively. (B) Co-evolving residues at the interface of VirB10 and VirB8.The ten top-scoring residue pairs in Dataset EV1 are mapped onto the corresponding Alphafold model. Colour coding for proteins is as in Fig. 1. Residue pairs are shown by red bars between their Calpha atoms. (C) Co-evolving residues at the interface of VirB10 and VirB6. The ten top-scoring residue pairs in Dataset EV1 are mapped onto the corresponding Alphafold model presented below. Colour coding for proteins is as in Fig. 1. Residue pairs are shown by yellow bars between their Calpha atoms. (D) Co-evolving residues at the interface of VirB10 with VirB3 and VirB4. The ten top-scoring residue pairs in Dataset EV1 are mapped for each interaction onto the corresponding Alphafold model. Colour coding for proteins is as in Fig. 1. Residue pairs are shown by bars between their Calpha atoms, colour-coded red, yellow and orange for VirB3-VirB10$_{NT}$, VirB4$_{central}$-VirB10$_{NT}$ and VirB4$_{outside}$-VirB10$_{NT}$ interactions, respectively.

