## [Peer Review File · The EMBO Journal]

Cryo-EM structure of a conjugative type IV secretion system suggests a molecular switch regulating pilus biogenesis

Gabriel Waksman and Kevin MACE

Corresponding authors: Gabriel Waksman (g.waksman@bbk.ac.uk) , Kevin MACE (kevin.mace@univ-rennes.fr)

Review Timeline:

Submission Date:	7th Feb 24
Editorial Decision:	11th Mar 24
Revision Received:	24th Apr 24
Editorial Decision:	16th May 24
Revision Received:	17th May 24
Accepted:	21st May 24

Editor: Ieva Gailite

Transaction Report:

Dear Gabriel,

Thank you for submitting your manuscript for consideration by the EMBO Journal. We have now received comments from three reviewers, which are included below for your information.

As you will see from the reports, all reviewers find the study of strong interest, and indicate only minor concerns that would have to be addressed in the revised manuscript. Based on these very positive assessments, I would like to invite you to revise the manuscript in response to reviewers' comments.

We generally allow three months as standard revision time. As a matter of policy, competing manuscripts published during this period will not negatively impact on our assessment of the conceptual advance presented by your study. However, please contact me as soon as possible upon publication of any related work to discuss the appropriate course of action. Should you foresee a problem in meeting this deadline, please let us know in advance to discuss an extension.

When preparing your letter of response to the referees' comments, please bear in mind that this will form part of the Review Process File and will therefore be available online to the community. For more details on our Transparent Editorial Process, please visit our website: <https://www.embopress.org/page/journal/14602075/authorguide#transparentprocess>. Please also see the attached instructions for further guidelines on preparation of the revised manuscript.

Please feel free to contact me if you have any further questions regarding the revision. Thank you for the opportunity to consider your work for publication.

With best wishes,

Ieva

Ieva Gailite, PhD
Senior Scientific Editor
The EMBO Journal
Meyerohofstrasse 1
D-69117 Heidelberg
Tel: +4962218891309
i.gailite@embojournal.org

We realize that it is difficult to revise to a specific deadline. In the interest of protecting the conceptual advance provided by the work, we recommend a revision within 3 months (9th Jun 2024). Please discuss the revision progress ahead of this time with the editor if you require more time to complete the revisions.

Referee #1:

In this manuscript by Mace and Waksman, the authors provide a more detailed structural understanding of the organization of the conjugative Type IV Secretion System (T4SS). Using larger datasets and continuing improvements in cryo-EM imaging processing algorithms, as well as improvements in structure prediction methods allow the authors to present a more detailed model this T4SS - including tracing symmetry mismatches, identifying new components, and proposing the identities of proteins found in regions of the cryo-EM density that remains at mid-resolution. This work represents an important advance from the previous published structure and provides interesting new models for the field. This is a complicated molecular machine to visualize, determine structures, and build models into. The authors should be commended for their mostly successful attempts at making the maps, models, and predicted structures clear and easy to follow. My one main comment is that Figure 6, which importantly shows how TrwE/Virb10 is a central component of multiple regions of the T4SS, would be easier to interpret if either in the figure (or as a supplementary figure) cryo-EM densities used to build the models for this proteins could somehow be included for each of the VirB10 models shown in panel B.

Minor Comment:

(Page 7): The conservation of this mismatch symmetry across various T4SS types emphasizes its importance. This distinct pattern observed in the OMCC, characterized by a quasi-symmetrical mismatch, suggests the existence of a specific maturation process during OMCC assembly, although the exact mechanism remains enigmatic. The authors suggest a plausible role for this asymmetry: to expand the dimensions of the I-layer structure, thereby achieving the ideal diameter to accommodate the growing pilus at a later stage.

It is unclear what "various T4SS types" are being referred to and which "authors" are being cited in the below excerpt. References and more precise language here would be very helpful.

Referee #2:

In the manuscript "Cryo-EM structure of a conjugative T4SS identifies a molecular switch regulating pilus biogenesis," K. Mace and G. Waksman report improved cryo-EM structures of *Agrobacterium tumefaciens* R388 secretion system (T4SS).

By collecting images of nearly 2 million particles, they were able to provide important new structural information on the molecular architecture of the R388 system. They provide new information on all the four major compartments: the outer membrane core complex (OMCC), the stalk, the arches, and the inner membrane complex (IMC).

In the OMCC the VirB9 helix, connecting the O-layer to the I-layer, is clearly visible, including side chains, even before applying symmetry. Moreover, after classification, they were able to capture three different conformations of the I-layer.

They solved the structure of the stalk tip formed by VirB5 that in their previously published map was not well resolved. This supported a very interesting hypothesis that the tip may penetrate target membranes.

They also discovered a fourth VirB8 molecule in the arches region and portions of the VirB10 region bound to the arches and to the IMC forming protein VirB6 and VirB4.

Since they found VirB10 occupying the VirB2 binding region on VirB6, the authors speculate a very interesting possible regulatory function of VirB10 on pilus biogenesis.

Overall the work presents a number of very interesting new details about this important and amazing molecular machine that will advance the field significantly.

Major issues:

The authors show coevolutionary matrices but do not provide any information on what software they used. What are the units of the colormap (Fig. 3C, 4B, 5D, 5E)? The author should also report the identified coevolving residues pairs. Are they modelled at the protein-protein interface in the AlphaFold model?

The authors should also report the PAE, and the ipTM for the AlphaFold models so readers can know how confident the oligomeric predictions are.

The MolA-C nomenclature is confusing - try "VirBAperi or VirBperiA or VirBperi1

The statement "we have recently shown that the presence of recipient cell considerably increases pilus biogenesis in the R388 system (AK Vaddakepat, Ho, Waksman, unpublished)" should not technically be made in the Results section without showing the data here.

Minor issues:

Despite the importance of having identified a new interaction between VirB6 and VirB10, further evidence should be obtained before claiming as a title that they identified a molecular switch. A slightly modified title such as "...a molecular switch that likely regulates pilus biogenesis" would be fine.

While the difference is very small, the authors technically shouldn't use "electron density map" since cryo-EM maps display Coulombic potential, not electron densities.

The words "superimposing entity" are puzzling - try "reference."

Referee #3:

This manuscript provides updated information on the structure of the R388 conjugative T4SS that has been revealed through on-going efforts in data collection and processing. The updates are substantial and will be of high value to the field of researchers trying to glean functional mechanism from these structural details. Perhaps most significant, the new analysis has revealed more density for the VirB10 protein which spans both the inner and outer membranes of the bacterium and appears to have several points of contact with the central stalk and pilus.

The authors do an excellent job of presenting the new structural findings with text and figures that allow the reader to appreciate the overall context. There is also a highly rigorous presentation of data quality through tables and density maps. I do not have any concerns and would like to congratulate the authors on a another significant advance for the T4SS field.

We are pleased that “all reviewers find the study of strong interest and indicate only minor concerns that would have to be addressed in the revised manuscript”.

You'll find below a detailed response to all reviewers' comments.

Referee #1

This referee found that our work represents an important advance from the previous published structure and provides interesting new models for the field. She/He remarks on the technical complexity of the work and, for the most part, on the clarity of our presentation.

Comment 1. The referee states: “My one main comment is that Figure 6, which importantly shows how TrwE/VirB10 is a central component of multiple regions of the T4SS, would be easier to interpret if either in the figure (or as a supplementary figure) cryo-EM densities used to build the models for this protein could somehow be included for each of the VirB10 models shown in panel B.

Response: We are a little unclear as to how to respond to this comment. We understand (perhaps wrongly) that the reviewer is recommending that we add the density for the various parts of VirB10 that we

identified in this study i.e. $\text{VirB10}_{\text{OM}}$, $\text{VirB10}_{\text{O-layer}}$, $\text{VirB10}_{\text{I-layer}}$, $\text{VirB10}_{\text{Arches}}$, $\text{VirB10}_{\text{IM}}$ and $\text{VirB10}_{\text{cyto}}$. However, the densities for $\text{VirB10}_{\text{OM}}$, $\text{VirB10}_{\text{O-layer}}$, $\text{VirB10}_{\text{I-layer}}$ are not new and have been reported before (Macé et al. Nature, 2022) while the densities for $\text{VirB10}_{\text{Arches}}$, $\text{VirB10}_{\text{IM}}$ and $\text{VirB10}_{\text{cyto}}$ are new and are all shown in Figure 1. So we don't really understand why we would be asked to show the densities again. So we have left Figure 6 as is.

Comment 2: The referee asked us to be more precise by listing the T4SSs we mention and add references on page 7 where we discuss symmetry mismatch in T4SSs.

Response: We agree with the reviewer that this paragraph requires clarification. However, we don't think it is necessary to list all the T4SSs with mismatch symmetry as it will make the reading quite difficult. Instead, as suggested by the reviewer, we have added all references where mismatch symmetry has been found in T4SS core complexes. As to the word "authors", since we were referring to us, we now say "we would like to suggest".

Referee #2

The reviewer states that, overall, the work presents a number of very interesting new details about this important and amazing molecular machine that will advance the field significantly. He/she has however a number of comments, which we have addressed.

Comment 1: The authors show coevolutionary matrices but do not provide any information on what software they used.

Response: The software used to generate the matrices is already mentioned in the Materials and Methods in the "structure validation" section. However, since the reviewer has numerous comments on the validation part of our manuscript, we sought of clarify the materials and methods section entitled "Structure validation using AlphaFold". We have added the following text: "To validate the new structural features and interactions presented in this study, we utilised AlphaFold (Jumper *et al.*, 2021) through the ColabFold advanced notebook (Mirdita *et al.*, 2022). This version of ColabFold generates various outputs, including distograms, PAE plots and ipTM scores (presented in Figures 3, 4 and 5) and lists of co-evolution pairs with their scores (presented in Table EV2 and Figure EV3)".

We have indeed added two Expanded View materials: Table EV2 that lists co-evolving residue pairs with score above the arbitrary threshold of 0.1 and Figure EV3 that maps for each interaction the

10 top-scoring co-evolving residue pairs for each interaction. This covers all requests from Reviewer 2 as detailed below.

Comment 2: What are the units of the colormap (Fig. 3C, 4B, 5D, 5E)?

Response: We now have added the units in the colormaps presented in Fig. 3C, 4B, 5D, 5E.

Comment 3: The author should also report the identified coevolving residues pairs. Are they modelled at the protein-protein interface in the AlphaFold model?

Response: All co-evolving residues pairs with scores > 0.1 are now listed in Table EV2 for each interaction. In Figure EV3, we also present the corresponding AlphaFold models with the 10 top-scoring co-evolving residue pairs mapped.

Comment 4: The authors should also report the PAE, and the ipTM for the AlphaFold models so readers can know how confident the oligomeric predictions are.

Response: All PAE plots and ipTM scores are now reported in the corresponding figures.

Comment 5: The MolA-C nomenclature is confusing - try "VirBAperi or VirBperiA or VirBperi1.

Response: This nomenclature was used in Macé et al Nature 2022 and we would like to keep it so as to not add more confusion and be consistent with our previous paper.

Comment 6: The statement "we have recently shown that the presence of recipient cell considerably increases pilus biogenesis in the R388 system (AK Vaddakepat, Ho, Waksman, unpublished)" should not technically be made in the Results section without showing the data here.

Response: The entire section is entitled "Results and Discussion". So, the statement may not need to be moved elsewhere.

Comment 7: Despite the importance of having identified a new interaction between VirB6 and VirB10, further evidence should be obtained before claiming as a title that they identified a molecular switch. A slightly modified title such as "...a molecular switch that likely regulates pilus biogenesis" would be fine.

Response: "Technically", the reviewer is right. So we now use the word "suggests" rather than "reveals".

Comment 8: While the difference is very small, the authors technically shouldn't use "electron density map" since cryo-EM maps display Coulombic potential, not electron densities. The words "superimposing entity" are puzzling - try "reference."

Response: We now use "EM density" and "reference".

Referee #3

This referee does not have any concerns and would like to congratulate the authors on another significant advance for the T4SS field.

We would like to thank the reviewers for their contribution to the improvement of the manuscript. We have now responded to all their comments and believe that the manuscript is now acceptable for publication in EMBO Journal.

Yours sincerely,

Gabriel Waksman

Dear Gabriel,

Thank you for submitting a revised version of your manuscript. I have now looked into your response to the reviewers' comments, and I find it reasonable. Therefore, I would be happy to extend formal acceptance of the manuscript once the editorial points below are addressed:

1. Please submit up to five keywords.
2. We require institutional email addresses for co-corresponding authors; it is currently missing for Kevin Mace.
3. Please add EV figure legends to the manuscript text file after main figure legends.
4. Please make sure that the funding information, including the grant numbers, is correct and identical both in the manuscript and our online system.
5. Please rename Table EV2 into Dataset EV1.
6. There is a callout left for a Table S1 - does it refer to Table EV1?
7. Please mention individual panels of Figure 1 in the manuscript text.
8. There is a reference to "Data not shown" on page 12. Such references are not allowed according to EMBO Press policy (see also <https://www.embopress.org/page/journal/14602075/authorguide#unpublisheddata>). Does this reference perhaps refer to your recent preprint (<https://www.biorxiv.org/content/10.1101/2024.03.04.583355v1>)? We do allow references to preprints in the following format; in the text: (preprint: NAME1 et al, YEAR); in the reference list: Author NAME1, Author NAME2 (YEAR) article title. bioRxiv doi [PREPRINT].
9. According to our updated Author guidelines (<https://www.embopress.org/competing-interests?=#disclosure-statement>), please state in the "Disclosure Statement & Competing Interests" section (replaces the "Conflict of Interest" section) that Dr. Waksman is a member of the editorial advisory board at The EMBO Journal. Our recommended formulation is as follows: "Gabriel Waksman is an editorial advisory board member at The EMBO Journal. This has no bearing on the editorial consideration of this article for publication."
10. I would like to suggest minor edits in the manuscript title, abstract and synopsis. I have also written a short blurb that will accompany the title of your manuscript in our online system. Please take a look at the text below and in the attached manuscript text file and let me know if any corrections are needed:

Title:

Cryo-EM structure of a conjugative type IV secretion system suggests a molecular switch regulating pilus biogenesis

Blurb:

New structural information provides insights into the four major compartments of a conjugative T4SS: the outer membrane core complex (OMCC), the stalk, the arches, and the inner membrane complex.

Synopsis:

Conjugative type IV secretion systems (T4SS) mediate transfer of genetic information between bacterial cells. This study presents a cryo-EM structure of a conjugative type IV secretion that provides insights into the various sub-complexes that form its machinery.

- New structural information provides insights into outer membrane core complex (OMCC) dynamics and mismatch symmetry.
- The complete structure of VirB8 is determined and a fourth VirB8 subunit is characterised within the arches and the inner membrane complex (IMC).
- The VirB10 protein structure from its N-terminus in the cytoplasm to its C-terminus in the OMCC is elucidated.
- Obstruction of the VirB2 pilin recruitment site on VirB6 by a trans-membrane region of VirB10 suggests a mechanism of pilus biogenesis regulation.

With best wishes,

Ieva

Ieva Gailite, PhD
Senior Scientific Editor
The EMBO Journal
Meyerhofstrasse 1

D-69117 Heidelberg
Tel: +4962218891309
i.gailite@embojournal.org

We realize that it is difficult to revise to a specific deadline. In the interest of protecting the conceptual advance provided by the work, we recommend a revision within 3 months (14th Aug 2024). Please discuss the revision progress ahead of this time with the editor if you require more time to complete the revisions.

The authors addressed the remaining minor editorial issues.

Dear Gabriel,

Thank you for addressing the final editorial points. I am now pleased to inform you that your manuscript has been accepted for publication in the EMBO Journal. Congratulations on a beautiful study!

If you have any questions, please do not hesitate to contact the Editorial Office. Thank you for your contribution to The EMBO Journal and congratulations on a great paper!

With best wishes,

Ieva
